# Effect of Al Content in Magnesium Alloy on Microstructure and Mechanical Properties of Laser-Welded Mg/Ti Dissimilar Joints

**DOI:** 10.3390/ma13122743

**Published:** 2020-06-17

**Authors:** Wen Dong, Rongrong Huang, Hongyun Zhao, Xiangtao Gong, Bo Chen, Caiwang Tan

**Affiliations:** 1Changchun Institute of Technology, Changchun 130000, China; 19b909121@stu.hit.edu.cn; 2State Key Laboratory of Advanced Welding and Joining, Harbin Institute of Technology, Harbin 150001, China; 19s130339@stu.hit.edu.cn (R.H.); zhaohy6691@hit.edu.cn (H.Z.); 3Shandong Provincial Key Laboratory of Special Welding Technology, Harbin Institute of Technology at Weihai, Weihai 264209, China; chenb@hit.edu.cn; 4Department of Mechanical and Aerospace Engineering, Missouri University of Science and Technology, Rolla, MO 65401, USA; xgxfv@umsystem.edu

**Keywords:** Mg/Ti dissimilar joints, laser welding, microstructure, interface reaction, mechanical properties

## Abstract

Laser penetration welding of magnesium alloys and pure titanium TA2 with unequal thickness was performed. Mg base metal with different Al content (AZ31B, AZ61A, AZ91D) was used to investigate the influence of Al element in microstructure and mechanical properties of Mg/Ti dissimilar joints. The results revealed that the change of Mg base metal did not influence the weld appearance of the joints. Three kinds of joint all presented the best mechanical property when the laser power was 3500 W. With the increase content of Al elements in Mg base metal, a reaction layer was observed which was identified as Ti_3_Al. The highest enrichment of Al element was obtained and its fraction reached 19.31 at% at the AZ91/TA2 interface. The chemical potential gradient of Al from AZ91 to Ti alloy was higher than that from the other two base metals based on thermodynamic calculation. The maximum fracture load reached 3597 N when AZ61 was employed as the base metal and the fracture position was the Ti base metal. AZ31/TA2 joints failed at the weld seam without necking due to the rapid propagation of cracks at the Mg/Ti interface. The AZ91/TA2 joint failed inside the Mg fusion zone with necking at the middle area of the weld, which resulted from the precipitation of brittle phases such as Mg–Al, Ti–Al phases in the fusion zone of Mg alloys.

## 1. Introduction

Nowadays, hybrid components of lightweight materials have been receiving extensive attention to achieve reduction of emissions and green development, which increases the demand for weight reduction in advanced manufacturing industry [1,2,3,4]. Magnesium (Mg), as one of the lightest engineering structural materials, has attracted wide attention due to its advantages of high strength, excellent formability and recyclability [5,6,7]. Titanium (Ti) has many features, such as high strength-to-weight ratio, good impact toughness and corrosion resistance [8,9]. The application of Mg/Ti dissimilar metal in spacecraft, such as the tail cabin and engine bracket, is of particular interest since the hybrid component would further reduce overall weight. Therefore, it is urgent to achieve reliable joining of Mg/Ti dissimilar metals [10,11].

However, achieving reliable bonding between Mg/Ti alloys needs to overcome the huge differences in theirs physical, chemical and metallurgical properties. No reaction layer or little solid solubility occurs, due to their immiscibility characteristics. The huge differences between the melting point of Mg (649 °C) and Ti (1678 °C) lead to severe burning and evaporation of the Mg alloy if using a traditional welding process. Besides, residual stress and deformation easily appear in the Mg/Ti dissimilar joints. In response to the above problems, investigations have been carried out and the results suggested that metallurgical reaction could be achieved by adding alloying elements such as Al, Cu, Ni [12,13,14,15] which can react with or possess obvious solid solubility in Mg and Ti alloy.

As for the influence of Al element on Mg/Ti welding, Cao et al. [16] carried out welding-brazing of TA2 and AZ31B using cold metal transfer (CMT) technique and found that the diffusion behavior of Al element in Mg alloy base metal and filler metal at the interface was an important factor contributing to the reliable joining of the Mg/Ti joint. Hu et al. [17] performed Mg/Ti laser welding-brazing with pure Al wire and found that a compound layer with the thickness of 1–2 μm was formed at the Mg/Ti interface. Energy-dispersive X-ray spectrometry (EDS) results showed that the main elements consisted of Mg, Al and Zn. Tan et al. [18] investigated the diffusion behavior of Mg–Al–Ti ternary system by thermodynamic calculation. It was revealed that Al element has a lower chemical potential on the Ti side, which promoted the uphill diffusion from the Mg alloy base metal to titanium alloy at the interface. Al–Ti intermetallic compound had a higher driving force of precipitation during the reaction process. It was concluded from the above results that Al element could not only have a large proportion of mutual solubility with Mg alloy, but also interact with Ti alloys to generate interfacial reaction layer, which played an important role in regulating Mg/Ti dissimilar welding. Moreover, the addition method and content of Al element also affect the weldability of Mg/Ti joints. When the Al element was fed into the molten pool through filler metal, the joint strength using AZ91 filler was greatly improved compared with AZ31 filler wire under various experimental parameters [18,19]. Zang et al. [20] the Al content accurately by changing the thickness of the Al interlayer. They found that the thickness of TiAl3 did not change obviously with the increasing Al interlayer thickness. However, the intermetallic compound became denser in the Mg fusion zone which damaged the mechanical property of Mg/Ti joints. The effect of Al on welding of Mg to Ti was also noticed in the friction stir welding (FSW) process. Aonuma et al. [21] performed FSW welding of pure titanium to different Mg sheets (AZ31B, AZ61A and AZ91D) to explore the effect of Al content on the interfacial microstructure of Mg/Ti joints. The diffusion of Al element was found to be different due to different content in Mg base metal. The thickness of the Ti–Al compound layer increased and the mechanical property declined with the increase of Al content.

At present, welding-brazing method was widely employed on Mg/Ti dissimilar metals while researches on deep penetration welding of thick plates were rarely reported. To the best of our knowledge, there are no investigations about influence of Al element as alloying element from Mg base metal on interfacial reaction and mechanical properties of laser welded Mg/Ti joints. In this study, laser deep penetration welding of Mg/Ti with unequal thickness was performed. Different Mg base metals (AZ31B, AZ61A, AZ91D) were used to join TA2. The purpose of this work was to establish the relationship between Al content and interfacial microstructure and thermodynamics, which lay a good foundation for the selection of the suitable base metal in the actual applications of Mg/Ti laser penetration welding with unequal thickness.

## 2. Materials and Methods 

In this work, three types of magnesium alloys with different Al contents (AZ31B, AZ61A, AZ91D) were selected as base metals in dimension of 100 mm × 50 mm × 3 mm. The base metal of titanium was TA2 with the dimensions of 100 mm × 50 mm × 1 mm. The chemical composition of the base metal was shown in Table 1. A 6 kW fiber laser (IPG-6000) (IPG Photonics Corporation, Oxford, MA, USA) with 1070 nm wavelength and KUKA KR60HA (KUKA, Augsburg, Germany) six-axis robot were used for welding. A Precitec YW52 (Precitec, Baden-Baden, Germany) welding head was employed with a linear design and a focal length of 300 mm. The focused spot size was 0.4 mm. Figure 1 shows the schematic of laser welding of Mg/Ti with unequal thickness. The assembly was adopted in a butt configuration with the gap of 0 mm. In order to ensure the full melting of the thick magnesium alloy, laser beam was irradiated on the surface of the Mg vertically and the laser offset distance to Mg was 0.3 mm. The defocused distance was selected as −1.5 mm with the beam spot dimension of 0.67 mm in order to avoid a large amount of burning of the Mg base metal. Ar gas was blown behind the laser beam as the shielding gas and under the weld seam to conduct root protection with the flow rate of 15 L/min to protect molten pool from oxidation. After several preliminary trials, the laser power adopted was in the range of 2.5–5.0 kW matching the welding speed of 1.5 m/min. The welding parameters employed in this work were shown in Table 2. The joints with the best performance were selected for further detailed analysis through comparing the weld appearances and mechanical property of the samples with different laser powers. During the welding process, 1 mm-thick shim was adopted on the thin Ti side to ensure that the thickness of the base metal was centered. The experiments performed with each laser power and Mg base metal were repeated three times.

Mg and Ti were easy to oxidize in the air due to active chemical properties. Before the experiment, mechanical and chemical cleaning were undertaken to remove the surface oxide film. Ti sheets was cleaned in the acid solution with the composition of 5% HF + 15% HCl + 80% distilled water and magnesium alloy was polished with a grinder and sandpaper. After the laser-welding process, typical areas of specimens with the dimension of 20 mm × 10 mm were cut to observe the cross section morphology of the joints. The samples were etched in acid solution (5.5 g picric acid, 2 mL acetic acid, 90 mL absolute alcohol and 10 ml distilled water) for 15 s to obtain the fusion line on the Mg side. The morphology of the sample was observed with an ultra-depth optical microscope (OLMPUS-DSX510) (OLMPUS Corporation, Beijing, China) and scanning electron microscope (SEM) with EDS (Carl Zeiss AG, Jena, Germany). The welded samples were cut perpendicular to the weld direction with a dimension of 10 mm × 80 mm for the tensile test. An electronic tensile tester (INSTRON MODEL 5967) (INSTRON CORPORATION, Shanghai, China) was used for the tensile test with a speed of 1 mm/min at room temperature according to the GB/T 2651-2008 standard [22]. In this study, the maximum load was used to express the tensile property of the overall joint due to the different cross-sectional areas of fracture. Three tensile samples from the same welded plates were prepared for tensile testing. The average value of the maximum tensile load was taken and the error range was calculated. The fracture surface was observed and analyzed by SEM. 

In order to analyze the driving force of the diffusion behaviors of atoms, thermodynamic calculations were performed. In the theory of thermodynamics, the driving force of elemental diffusion is the chemical potential. The Miedema model and Toop model were adopted to analyze the diffusion and aggregation of Al and Ti. 

The Gibbs free energy of Mg–Al–Ti ternary system was calculated through the Toop model [23] which was an extended asymmetry model developed from Miedema model [24]. The Gibbs free energy and the chemical potential of elements were calculated through the following equations: (1)$$G^{E}=\frac{x_{2}}{1-x_{1}}G_{12}^{E}\left(x_{1},\;1-x_{1}\right)+\frac{x_{3}}{1-x_{1}}G_{13}^{E}\left(x_{1},\;1-x_{1}\right)+{\left(x_{2}+x_{3}\right)}^{2}G_{23}^{E}\left(\frac{x_{2}}{x_{2}+x_{3}},\frac{x_{3}}{x_{2}+x_{3}}\right)$$
(2)$$G_{m}=G^{ID}+G^{E}$$
(3)$$\mu_{i}=\frac{\partial G_{m}}{\partial x_{i}}$$
where $G^{E}$ and $G_{ij}^{E}$ represented the molar excess Gibbs energies.$\;G_{m}$ and $G^{ID}$ were Gibbs free energy and ideal solution approximation, respectively. $\mu_{i}$ represented chemical potential of element i and $x_{i}$ was the mole fraction of i.

In the binary system, $G_{ij}^{E}$ could be calculated through $\Delta H_{1,2}$ and the relationship between them is shown as follows:(4)G12E=ΔH1,2[1−(1Tm,1+1Tm,2)/14]
where ΔH was the formation enthalpy of binary system. $T_{m}$ was the melting point of components.

ΔH in the binary system could be calculated applying to the Miedema model.
(5)ΔH1,2=f1,2x1[1+μ1x2(φ1−φ2)]x2[1+μ2x1(φ2−φ1)]x1V123[1+μ1x2(φ1−φ2)]+x2V223[1+μ2x1(φ2−φ1)]
(6)f1,2=2pV123V223[q/p(ΔnWS1/3)2−(Δφ)2−α(r/p)](ΔnWS1/3)1−1+(ΔnWS1/3)2−1
where *φ* represented the electronegativity, V was the molar volume, $n_{ws}$ was the electron density, and *q*, *r*, *μ*, *a*, *p* were experimental constants. Physical parameters and values listed in Table 3. 

## 3. Results and Discussion

### 3.1. Joint Appearances and Cross Sections

Magnesium alloys were easy to evaporate and burn during the welding process due to their lower melting point (649 °C) and boiling point (1107 °C). In order to obtain high-quality Mg/Ti laser-welded joints, experiments were first conducted with different heat inputs to explore the optimal welding parameters. The stability of the welding process and the weld appearances without obvious defects were carried as the reference to evaluate the suitability of the welding parameters. It was found that the joint appearance of the welded seam did not change obviously when different Mg base metal was used under the same welding parameters. To avoid repetition, only one set of results was presented. Figure 2 showed representative appearances of AZ61/TA2 laser welded joints with unequal thickness produced at different laser powers. As shown in Figure 2a,b, the weld width was small due to the lower heat input employing laser power of 2.5 kW and 3.0 kW. Figure 2c–e indicated that the weld width increased with the increase of laser power. Besides, the increasing effect on the front side of the weld was small while the significant effect on the back side appeared. It was mainly attributed to the formation of the keyhole during the welding process due to the higher laser energy density acting on the weld. As the laser power increased, the penetration of the keyhole increased resulting in the rapid collapse of the front side of the molten metal and the increased fluidity of the molten metal on the back side. The back surface of the weld was rough because of the instability of the molten pool under the effect of gravity. When the laser power increased to 5 kW, molten metal collapsed seriously and over-burning defect was obtained as shown in Figure 2f. The molten pool spread poorly on the back surface of the joints.

Figure 3 showed cross-sectional morphology of Mg/Ti joints under different laser powers. During the welding process, both Mg alloy and TA2 were partially melted. The red line in Figure 3 was the fusion line. The entire joints could be divided into two parts: fusion zone of Mg and fusion zone of Ti. Due to the low melting point of Mg and the laser offset to the Mg side, the melting area of the Mg alloy was larger than that of Ti. The seam width and spread width at the corresponding position in Figure 3 were measured and the results are shown in Figure 4. Among them, the seam width was defined as the distance from the fusion line at Mg side to the left side of the Ti fusion zone and the spread width was defined as the distance from the right side of Mg alloys to the left side of the Ti fusion zone at the upper and lower surface. The schematic diagrams of the measurement location are shown in Figure 4. With the increase of laser power, the upper surface of welding seam gradually collapsed and the seam width at the upper face was larger than that at the lower surface. With the increase of laser power, the melting amount reflected from the seam width of magnesium alloy was increasing slightly on the upper surface (0.98 mm–2.27 mm) while significantly on the lower surface (1.3 mm–3.16 mm), which was consistent with the welding appearance. In addition, larger heat input improved the spread width of the melting metal on the Ti side at the lower surface (0.29 mm–1.6 mm) as shown in Figure 4b. The fusion zone on the Ti side appeared semicircular. The melting amount at the upper and lower side was significantly greater than the middle. During the welding process, two molten pool on the upper and lower sides of the Ti side existed with the keyhole as the boundary, which resulted in the heat input at the upper and lower side being greater than the middle position [25]. In summary, the quality of the weld appearance was an important factor in evaluating the welding quality of the joint. Uniform and sound appearance contributed to the improvement of joint strength. When the laser power was low, the melting amount of base metal was insufficient to achieve reliable joining while excessive heat input would limit the improvement of mechanical property of the joint due to the presence of the welding defects.

### 3.2. Mechanical Property

Tensile test was conducted to evaluate mechanical property of the Mg/Ti laser welded joints and the results are shown in Figure 5. The variation of joint strength produced with different Mg alloy base metal was found to present the similar trend with the increase of the laser power. Poor joint performance occurred in the case of low laser power. With the laser power increasing, the tensile load of the joints increased rapidly and reached the maximum at laser power of 3.5 kW. The maximum load was 3597 N of AZ61/TA2, 3346 N of AZ91/TA2 and 3114 N of AZ31/TA2. With a further increase of laser power the joint strength decreased, which was related to the collapse of the weld as shown in Figure 3. In order to further explore the influence of Al content on the quality of Mg/Ti deep penetration joints, joints with the highest fracture load which performed at laser power of 3.5 kW was selected to analyze the microstructure and fracture behaviors in the next sections.

### 3.3. Interfacial Microstructure

Figure 6 showed the interfacial microstructure of Mg/Ti laser welded joints with different Mg base metal using laser power of 3.5 kW. The observed locations were also indicated in the images, which were the upper, middle and lower parts of the joints. Figure 6a–c shows the interfacial morphology of the AZ31/TA2 joints, and no obvious interfacial reaction layer was observed in the upper and middle parts of the joints while a thin interfacial reaction layer appeared in the lower part. Compared with the AZ31/TA2 joint, an ultra-thin reaction layer grew from the surface of the Ti in the middle of the joints when the AZ61 base metal was employed as shown in Figure 6e. The feature was observed to display discontinuous and serrate-shaped morphology. When the Al content of the Mg alloy base metal continued to increase, a clear interface reaction layer could be seen in the middle and lower parts of the Mg/Ti interface. Compared with the other two joints, the thickness of the reaction layer in the middle parts of the AZ91/TA2 joints increased significantly. The formation of a reaction layer indicated that metallurgical bonding was achieved at Mg/Ti interface. The possibility of improved bonding between Mg and Ti by adding more Al content into Mg alloy base metal was confirmed. The reaction interfacial layer was mainly identified as the Ti_3_Al via transmission electron microscopy (TEM) (JEOL, Tokyo, Japan) analysis, which was reported in our previous study [19]. The EDS results at the interfacial reaction layer (P3) in this study (as shown in Table 4) also confirmed this finding. In addition, the precipitation of scattered white phase as Mg_17_Al_12_ was obviously observed at the fusion zone at Mg side for AZ61/TA2 and AZ91/TA2 joints. Zang et al. also obtained similar results [19].

The reaction layer at the Mg/Ti interface was thin and the morphology were blurred since small Al content in the Mg base metal was involved into the interfacial reaction [25]. Therefore, line scanning analysis across the Mg/Ti interface was performed through SEM-EDS testing to obtain the concentration profiles of the main elements in order to estimate the degree of interface reaction and the results are shown in Figure 7. On the whole, the Mg content of decreased while the Ti content increased at the interface. Al element segregation was observed at the interface of Mg/Ti indicating the appearance of atomic diffusion of Al from the Mg base metal to Mg/Ti interface. However, the enrichment degree of Al was different using different Mg alloy base metal. The enrichment of Al element was not obvious in the upper and middle regions for the AZ31/TA2 joints. While at the bottom of the joint, the atomic fraction of Al element at the interface reached 12.96 at%. When using AZ61 as the base metal, Al element was obviously enriched in the middle and lower parts of the joint and the highest Al content at the lower parts could reach 15.62 at%. With the Al content continually increased to 9 wt.%, Al significantly enriched at the whole regions along the interface compared with the other two joints and the highest Al content increased to 19.31 at% which indicated the great reaction during the metallurgical bonding of Mg and Ti.

### 3.4. Bonding Mechanism 

To clearly clarify the interfacial reaction at Mg-Ti interface, Gibbs free energy and chemical potential of each element (Mg, Al and Ti) at 300 K, 900 K, 1300 K, 1600 K and 2000 K were calculated and analyzed. These several temperatures were selected representing different stages, i.e., room temperature; Mg, Al and Ti elements remained at a solid state; Ti was at solid state while Mg and Al was melted; Ti was at solid state, Al existed at liquid form while gaseous Mg appeared; Ti started to be melted. Thermodynamic characteristics at these temperatures could help understand atomic diffusion and interfacial reaction during the cooling process. As shown in Figure 8a, Al element had the lowest chemical potential at Ti side in Mg–Ti–Al ternary system, indicating that the general diffusion tendency of Al atoms from Mg base metal side to Mg/Ti interface. The chemical potential gradient of Al from AZ91, AZ61 and AZ31 to Ti was −117.65 KJ/mol, −116.19 KJ/mol and −112.49 KJ/mol, respectively, suggesting that the driving force of Al from AZ91 to Ti substrate was a little higher than that from the other two base metals. It coincided with the fact that higher content of Al was detected at the AZ91/Ti interface shown in Figure 7. In addition, Al atoms could diffuse from Mg base metal (AZ91, AZ31 or AZ61) to Ti substrate in a wide range. Take AZ91–Ti system as example, Al originally diffused from AZ91 to Ti at Route 1 (shown in Figure 8a), and then continuously diffused to Al–contained Ti base metal at Route 2, Route 3, and even could diffuse to 0.75 at% Al–contained Ti alloy at Route 11. However, the maximum Al content at this work was 27.08 at% (EDS result in Table 4, and as marked in red circle in Figure 8a) due to the high cooling rate of laser welding, which provided a possible formation of Ti_3_Al phase. Ti atoms cannot diffuse to Mg base metal due to the positive chemical potential difference while they were strongly attracted by the Al atoms gathered at Mg/Ti interface, as shown in Figure 8b, which indicated that the Ti–Al phase was generated at the interface other than in the fusion zone. The Gibbs free energy of Mg–Al–Ti system at different temperature was also calculated, based on which possible formation phases could be estimated. Figure 8c summarizes the Ti/Al atom ratio at minimum Gibbs frees energy (the values were shown in Table 5) of 0.001 Mg–Al–Ti system (or at Ti base metal) and Figure 8d showed the Gibbs free energy at 900 K. At equilibrium state without limit of Al contents, TiAl phase was estimated to generate as shown in Figure 8c and Point 1 in Figure 8d. In this work, maximum Al content was 27.08 at% and the phase with Ti/Al atom ratio of 2.7 was estimated to form, which turned out to be Ti_3_Al phase (marked as Point 2 in Figure 8d). It also can be estimated in Figure 8d that Mg_17_Al_12_ was produced in the fusion zone. The Mg–Al eutectic phase with a similar constitution of 0.74 at% Mg and 0.25 at% Al has been detected in previous work [26].

Based on the above analysis, the bonding mechanism of Mg/Ti laser deep penetration welding was illuminated and the schematic diagram is shown in Figure 9. Firstly, laser beam irradiated on the Mg base metal directly. Mg alloys melted under the effect of high temperature. At the same time, laser energy was transferred to the Ti side and the thin TA2 sheets melted simultaneously. The atoms adjacent to the Mg/Ti interface were activated. Mg/Ti atoms did not interact with each other due to the immiscibility. Hence, the Mg and Ti atoms hardly diffused to each other in the early stage of the heating. Al atoms existing in the molten pool at Mg side started to diffuse to the Mg/Ti interface under the driving force with the decrease of chemical potential as shown in Figure 9a. A small amount of Mg atoms would diffuse to Al atoms due to the attractive effect of Mg and Al atoms. As base metal continued to melt, Al atoms enriched abundantly at the Mg/Ti interface. Ti atoms existing in the molten pool at Ti side started to diffuse to Mg/Ti interface under the attraction of the Al–enriched layer and the stirring effect of molten pool. At the same time, the increase of Ti fraction in the molten pool resulting from the continuous dissolution of TA2 would further accelerate the diffusion of Al atoms as shown in Figure 9b. As the temperature decreased, the increased amount of Al and Ti atoms at the front of the Mg/Ti interface provided convenient conditions to generate the Al/Ti intermetallic compound layer [27] as shown in Figure 9c. As the temperature further decreased, eutectic reaction occurred and structure consisted of α-Mg and Mg_17_Al_12_ compound formed at the fusion zone of Mg. In addition, with the condition of laser deep penetration welding, part of Ti atoms entered the fusion zone of Mg alloys with the stirring effect of the keyhole. These Ti atoms mixed with Mg and Mg–Al phase leading to the formation of the brittle phase in the fusion zone [21].

According to the above results, it was found that the interface reaction of Mg/Ti laser welding was related to the amount of activated Al atoms diffusing from Mg base metal to the interface of Mg/Ti, which was influenced by Al content of the Mg alloy base metal and specific welding process with various thermal cycles at the different interface positions. According to the results of line scanning, the segregation of Al at the upper interface was extremely weak, while the enrichment degree of Al atoms in the lower position was much higher than that in the upper or middle position in the same joints. This phenomenon was related to the stirring action of molten pool induced by keyhole during laser deep penetration welding process and the stirring effect of molten pool is shown in Figure 10. The stirring effect on the molten pool became stronger when approaching the keyhole. Therefore, the large size of the keyhole in the upper part of the joint prevented the full contact between the liquid Mg and Ti, which made it difficult for Al element in base metal to diffuse to the interface and produce an obvious reaction layer. The keyhole effect decreased along the plate thickness. With the depth of the keyhole increased, the penetration ability of the keyhole was weakened, which reduced the stirring effect of molten pool and provided more favorable conditions for the diffusion of the Al element. As a result, the Al element could fully diffuse to the interface resulting in a higher fraction of Al elements at the Mg/Ti interface in the lower part of the joints. Comparing joints using AZ31, AZ61 and AZ91 as the Mg base metal, it was found that with the increase of Al content in Mg alloy base metal, the interface reaction layer became more evident, especially in the middle and lower parts of the joints. This observation was in consistence with the results obtained in friction stir welding of Mg/Ti [21]. In this previous study, the increased Al content in Mg alloy base metal led to more activated Al atoms, which promote sound metallurgical reaction between Mg and Ti base metal.

### 3.5. Fracture Behaviors

In order to analyze the effect of three types of Mg base metal with different Al content on the mechanical properties of the weld, the load-displacement curve and fracture location at laser power of 3.5 kW was shown in Figure 11. It was found that the failure position of AZ31/TA2 and AZ91/TA2 joint was at weld seam. AZ31/TA2 joints failed without deformation significantly while certain necking on the Ti side was obtained during the fracture process of AZ91/TA2. AZ61/TA2 joints fractured at the Ti base metal, indicating good interfacial bonding occurred at the interface. The displacement generated during the tensile process of AZ61/TA2 joint was much greater than that of AZ31/TA2 and AZ91/TA2. 

Figure 12 shows the fracture behavior of AZ31/TA2 and AZ91/TA2 joints with the laser power of 3.5 kW. It can be seen from Figure 12a,b that fracture surface of AZ31/TA2 joints along the weld direction had two different areas: zone 1 and zone 3 characterized by a rough surface where a small amount of Mg in middle part of joints was observed. The fracture surface of zone 2 and zone 4 was smooth, where the cracks propagated rapidly along the Mg/Ti interface. The cross-sectional morphology showed that the joints failed entirely along the Mg/Ti interface. Compared with the AZ31/TA2 joint, the AZ91/TA2 joint did not completely fracture along the Mg/Ti interface. The residual Mg alloy was observed attached to the Ti surface in the middle region of the joint as shown in Figure 12e, which indicated that the crack propagated inside the Mg alloy at the middle part of joints instead of fracturing along the Mg/Ti interface during the tensile process. The fracture surface of AZ91/TA2 joints along the weld direction showed uniform morphology along the weld direction, suggesting that joints failed along the Mg/Ti interface at upper and bottom area while inside of the Mg alloy in the middle region. Combining the microstructure shown in Figure 6, when AZ31 was employed as base metal, a reliable intermetallic compound layer cannot be formed at the interface due to the low Al content, resulting in a low Mg/Ti interface strength. Therefore, cracks propagated and eventually failed along the interface in AZ31/TA2 joint. When AZ91 was used as base metal, the Ti–Al reaction layer was found at the interface, which improved the strength of the Mg/Ti interface compared with the AZ31/TA2 joints. In addition, Figure 12e revealed that linear Ti–Al structure existed in the fusion zone of Mg. A similar phenomenon was observed in previous research [25]. In heating and cooling process of laser welding, some Ti fragments were involved into the fusion zone and interacted with the Al element that precipitated from Mg (Al) solid solution, giving rise to the formation of the Ti–Al brittle phase and reducing the tensile property of the Mg fusion zone.

Figure 13 showed the SEM morphology of fracture surface of the AZ31/TA2 joint. Figure 13a,b presented areas 1 and 2 in Figure 11 at the Ti side, respectively. The feature of fracture surface at Ti side was characterized by cleavage steps and river-like patterns indicating the brittle fracture. At the Ti side, the fracture surface consisted of mixed structure of Ti–Al phase (26.88 at% Al, 67.72 at% Ti) and residual Mg (96.17% at% Mg, 3.71 at% Al, 0.12 at% Ti). Tan et al. [19] obtained similar results in that the serrate-shaped microstructure reaction layer was attached to the Ti substrate, which mainly resulted in the high resistance to crack propagation. Figure 13b shows the smooth and flat area where the crack quickly propagated. The fracture surface on the Ti side was mainly identified as the Ti–Al compound (12.02 at% Al, 81.13 at% Ti) with few residual strips of magnesium alloy (92.43% at% Mg, 5.44 at% Al, 2.13 at% Ti). The formation of the Ti–Al phase at the fracture surface confirmed that Al element had diffused to Mg/Ti interface. However, insufficient reaction of Mg/Ti due to the low Al content resulted in lower tensile load and joint toughness. Figure 13c,d show the fracture surface morphology on the Mg side of AZ31/TA2 joint, which mainly consisted of Mg alloy (97.29% at% Mg, 2.57 at% Al, 0.15 at% Ti). However, the Al content at the area where the cracks propagated quickly in Figure 13d exceeded that in the Mg base metal (10.82 at% Al, 89.09 at% Mg), indicating that part of the Al atoms still concentrated in the Mg alloy and did not participate in the interface reaction. The interface reaction was insufficient due to its low Al content at Mg/Ti interface, which resulted in the poor mechanical property of AZ31/TA2 joints.

Figure 14 shows SEM morphology of the AZ91/TA2 joint fracture on Ti side at the middle regions. In this region, the fracture surface completely consisted of Mg (94.02% at% Mg, 5.15 at% Al, 0.82 at% Ti) indicating that the joints failed inside the Mg alloy. Higher magnification exhibited typical morphology of mixed of cleavage fracture and ductile fracture. Nevertheless, little Ti–Al phase (19.26 at% Al, 74.37 at% Ti) was observed inside the Mg alloy which corresponded to Figure 12. The formation of brittle phases in the fusion zone of Mg alloy resulted in the weakest area of the joint transferring from the Mg/Ti interface to the fusion zone of Mg.

The main reason for the differences of the property was the diffusion and reaction behavior of the Al element during the welding process. As the Al content in Mg base metal increased, more Al atoms diffused to the interface and participated in the interfacial reaction, which was beneficial to the formation of a reliable reaction layer at the interface. Therefore, the interfacial bonding degree of the AZ31/TA2 joint was the weakest, resulting in quick propagation of a crack along the interface during the tensile process. However, the increasing quantity of Al content in AZ91 Mg alloys led to the formation of brittle phases such as Mg–Al, Ti–Al phases in the fusion zone of Mg, which increased the crack propagation and damaged the joint performance. The interfacial strength was enhanced while the property of the fusion zone was damaged for AZ91/TA2 joints, which was responsible for the transformation of the weakest position from the Mg/Ti interface to the fusion zone of Mg alloys at the middle region of joints. When AZ61 was employed as the base metal, the appropriate Al content not only ensured a reliable reaction layer at the Mg/Ti interface which improved the interfacial strength, but also did not seriously damage the performance of the fusion zone of the Mg alloy. Therefore, AZ61/TA2 joints had good performance as expected.

## 4. Conclusions

Laser deep penetration of Mg alloys with different Al content (AZ31B, AZ61A, AZ91D) and TA2 with unequal thickness was performed. The effect of Al content on weld appearances, cross-sectional morphology, microstructure of Mg/Ti interface and mechanical property was observed and analyzed. The following conclusions were drawn: With the increase of laser power, the front side of the weld seam gradually collapsed and the bead width on the back side increased. The change of the Mg base metal did not affect the joint appearance. The strength of three types of joints reached highest with laser power of 3.5 kW.An obvious segregation of Al elements was obtained. The enrichment degree of Al was promoted when Al content in the Mg base metal increased. The Al fraction reached 19.31 at% when AZ91 was employed. A reaction layer was obtained along the Mg/Ti interface indicating that metallurgical bonding of Mg/Ti was achieved and the degree of interfacial reaction improved with increasing Al content in the Mg base metal.The maximum fracture load reached 3597 N when AZ61 was used as the Mg base metal. Both the sound mechanical property of AZ61/Ti interface and fusion zone resulted in the failure at the Ti base metal.AZ31/TA2 joints failed along the Mg/Ti interface where the cracks rapidly propagated due to the poor interfacial bonding. AZ91/TA2 joints failed in the Mg fusion zone in the middle area which resulted from the improvement of interfacial strength and declining strength of the fusion zone.

## Figures and Tables

**Figure 1 materials-13-02743-f001:**
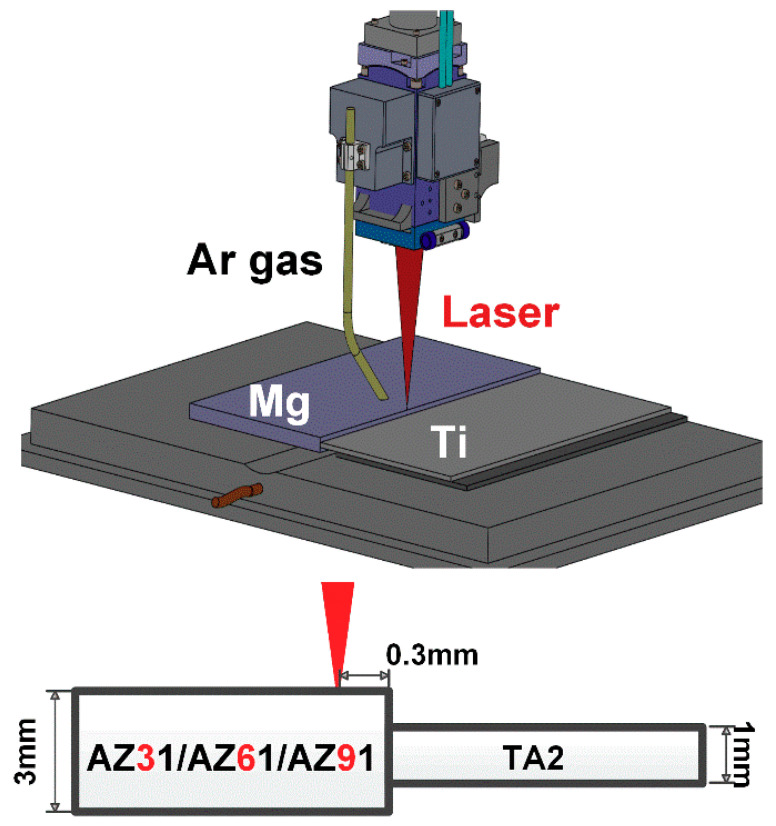
Schematic of laser deep penetration welding of Mg/Ti dissimilar joints.

**Figure 2 materials-13-02743-f002:**
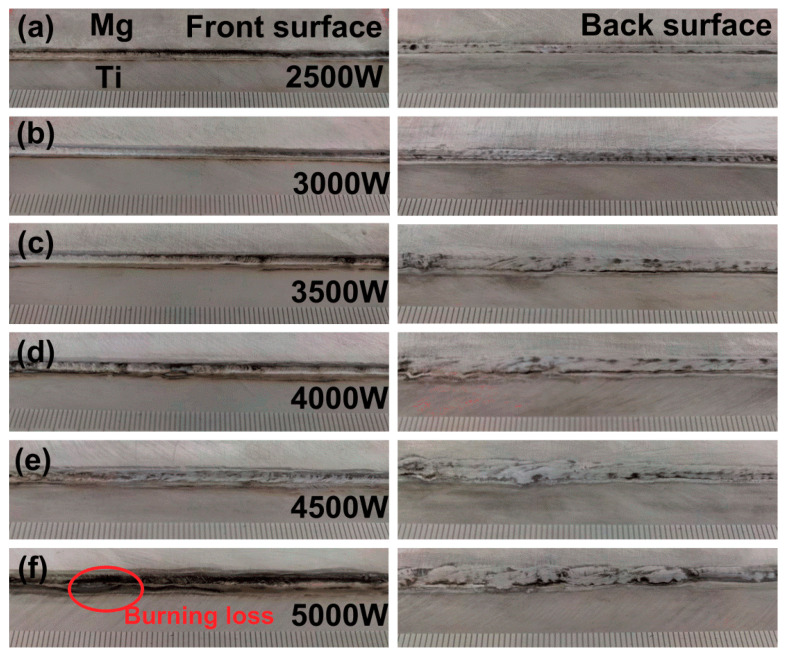
Full weld profile of laser deep penetration of AZ61/TA2 with variation of laser power: (**a**) 2.0 kW; (**b**) 2.5 kW; (**c**) 3.0 kW; (**d**) 3.5 kW; (**e**) 4.0 kW; (**f**) 5.0 kW.

**Figure 3 materials-13-02743-f003:**
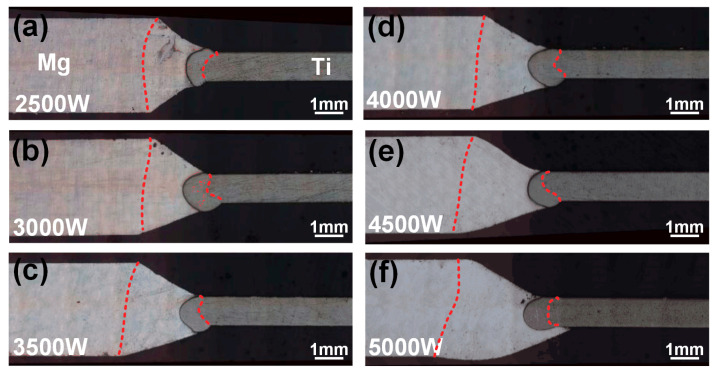
Cross sections morphology of laser deep penetration welding of AZ61/TA2 at various laser power: (**a**) 2.0 kW; (**b**) 2.5 kW; (**c**) 3.0 kW; (**d**) 3.5 kW; (**e**) 4.0 kW; (**f**) 5.0 kW.

**Figure 4 materials-13-02743-f004:**
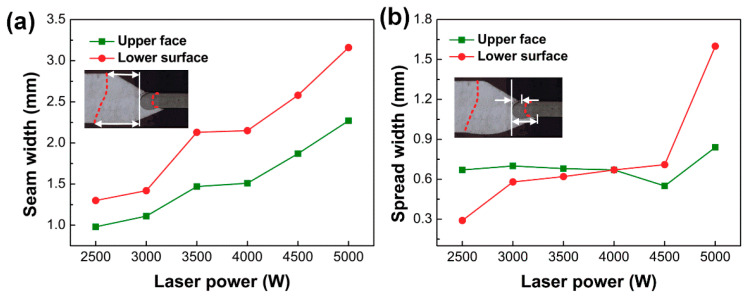
The position and results of measurement in Figure 3: (**a**) seam width; (**b**) spread width.

**Figure 5 materials-13-02743-f005:**
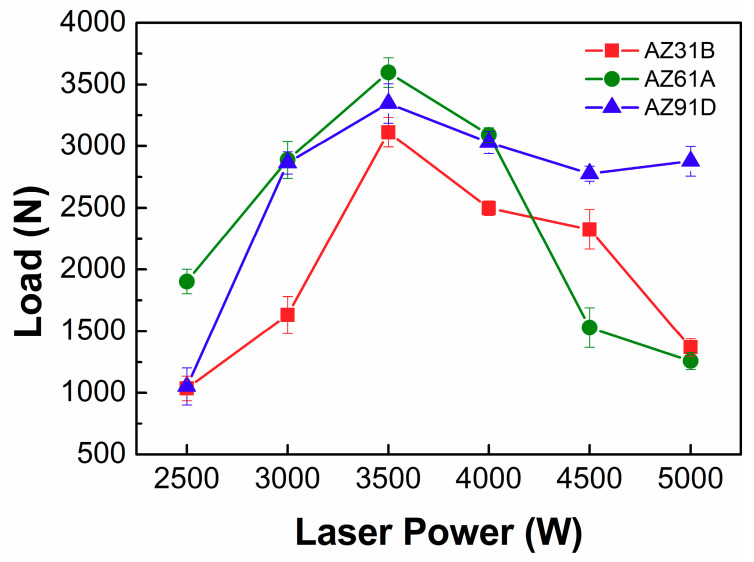
Fracture load of laser deep penetration welding of Mg/Ti at various laser powers.

**Figure 6 materials-13-02743-f006:**
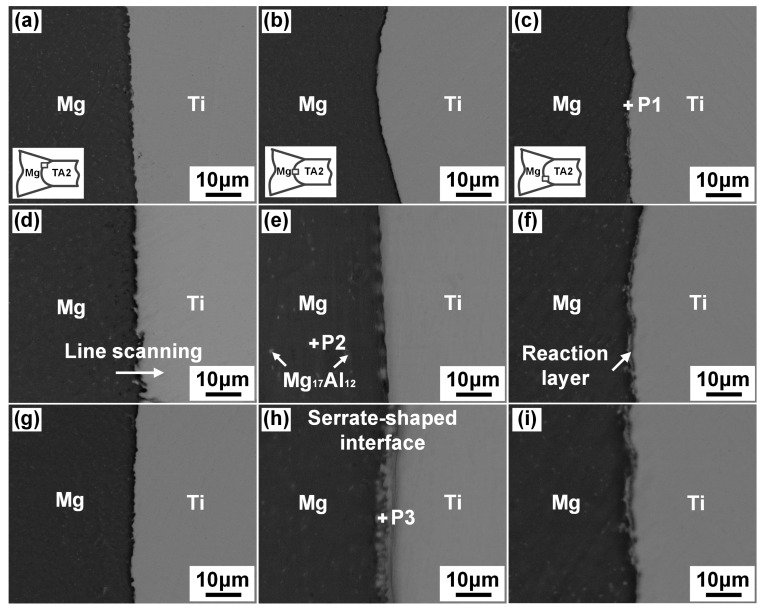
Interfacial microstructure of laser deep penetration welding of Mg/Ti at 3.5 kW: (**a**–**c**) AZ31/TA2; (**d**–**f**) AZ61/TA2; (**g**–**i**) AZ91/TA2.

**Figure 7 materials-13-02743-f007:**
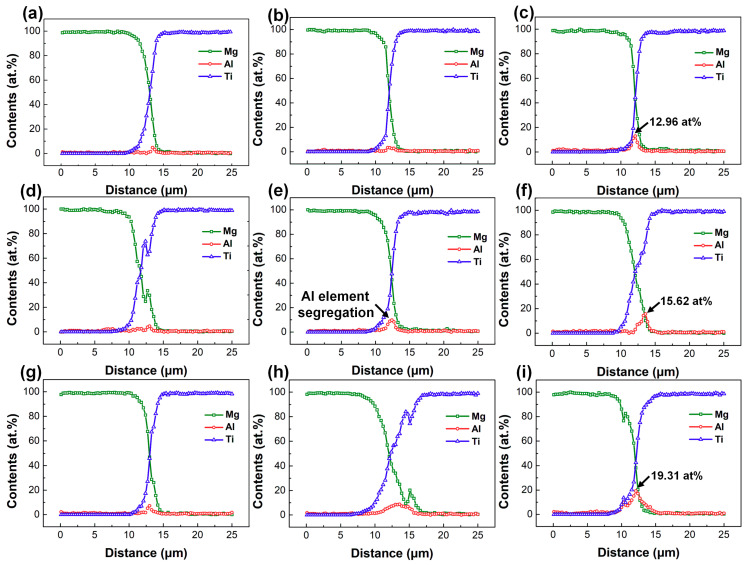
Line scanning results of Mg/Ti laser deep penetration welding joints at laser power of 3.5 kW: (**a**–**c**) AZ31/TA2; (**d**–**f**) AZ61/TA2; (**g**–**i**) AZ91/TA2.

**Figure 8 materials-13-02743-f008:**
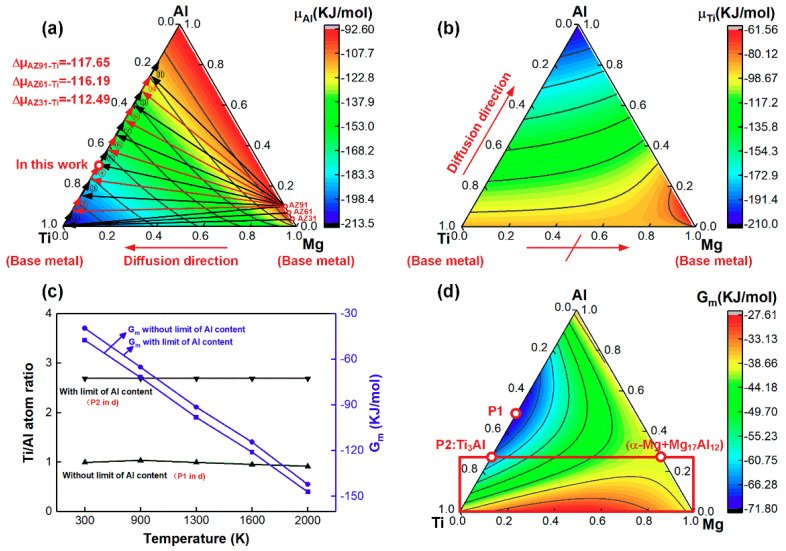
Thermodynamic calculation results: (**a**,**b**) chemical potential of Al and Ti at 1600 K; (**c**) Ti/Al atomic ratio at minimum Gibbs free energy of 0.001 Mg–Al–Ti system (or at Ti base metal) at different temperatures; (**d**) Gibbs free energy of Mg–Al–Ti system at 900 K.

**Figure 9 materials-13-02743-f009:**
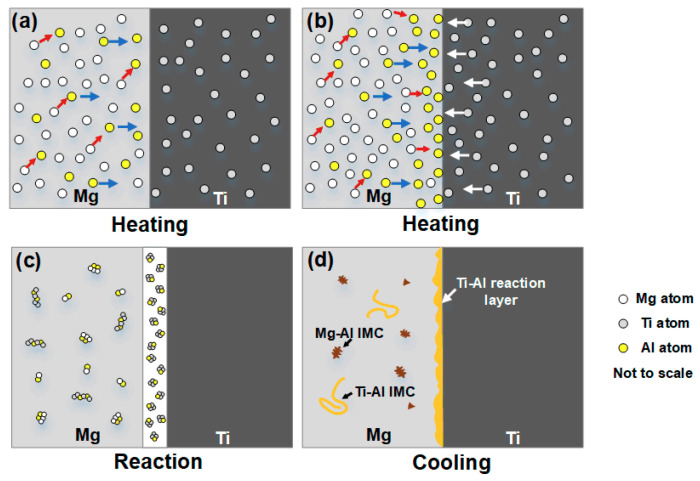
Reaction mechanism of laser deep penetration of Mg/Ti dissimilar joints: (**a**,**b**) process of welding heating; (**c**) process of atomic diffusion reaction; (**d**) process of cooling and compound formation.

**Figure 10 materials-13-02743-f010:**
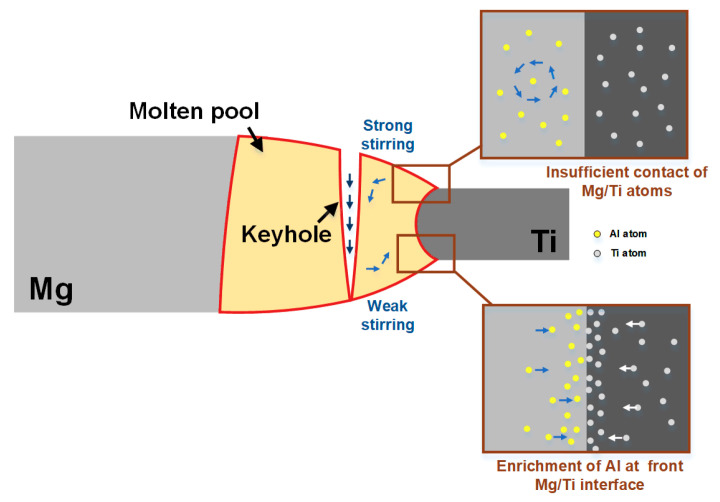
Schematic diagram of stirring effect of molten pool.

**Figure 11 materials-13-02743-f011:**
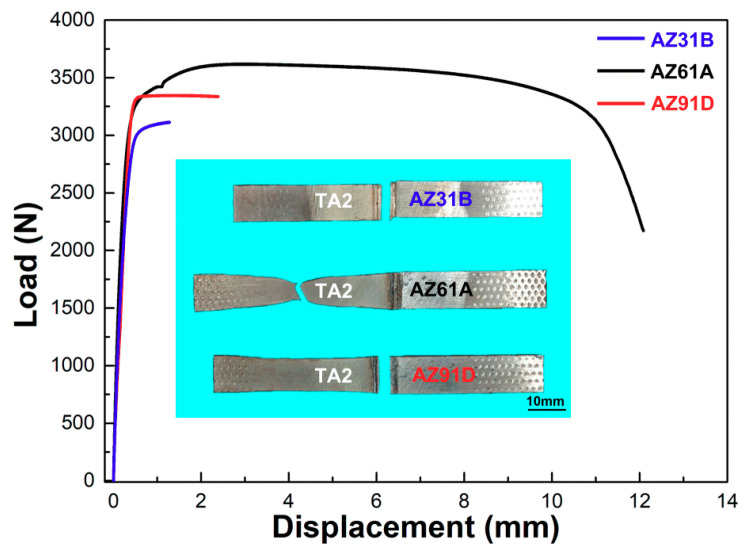
Load-displacement curves and fracture mode of laser deep penetration welding of Mg/Ti at lase power of 3.5 kW.

**Figure 12 materials-13-02743-f012:**
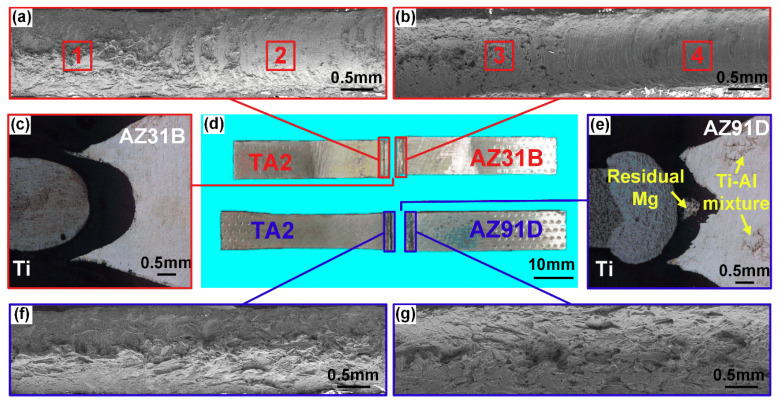
Fractography of Mg/Ti joints: (**a**,**b**) scanning electron microscope (SEM) images of fracture surface of TA2/AZ31B joints; (**c**) fracture path of TA2/AZ31B joints; (**d**) macro-morphology of TA2/AZ31B and TA2/AZ91D joints; (**e**) fracture path of TA2/AZ91D joints; (**f**,**g**) SEM images of fracture surface of TA2/AZ91D joints.

**Figure 13 materials-13-02743-f013:**
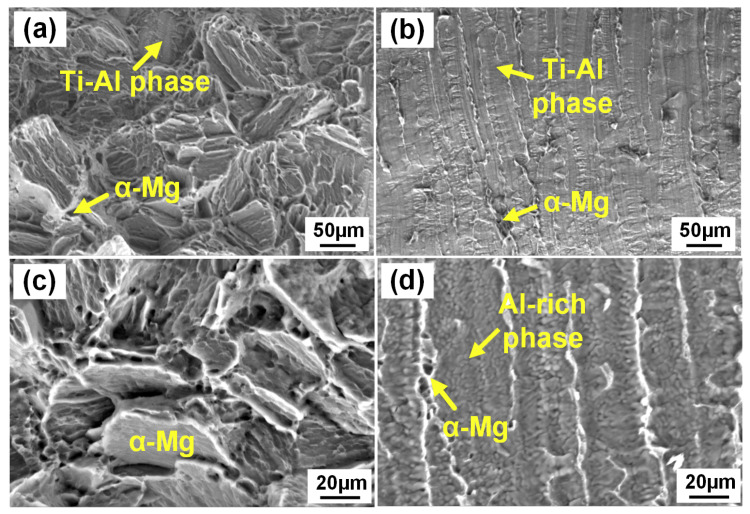
SEM morphology of fracture surface of AZ31/TA2 joints: (**a**) area 1 in Figure 9 at the Ti side; (**b**) area 2 in Figure 9 at the Ti side; (**c**) area 3 in Figure 9 at the Mg side; (**d**) area 4 in Figure 9 at the Mg side.

**Figure 14 materials-13-02743-f014:**
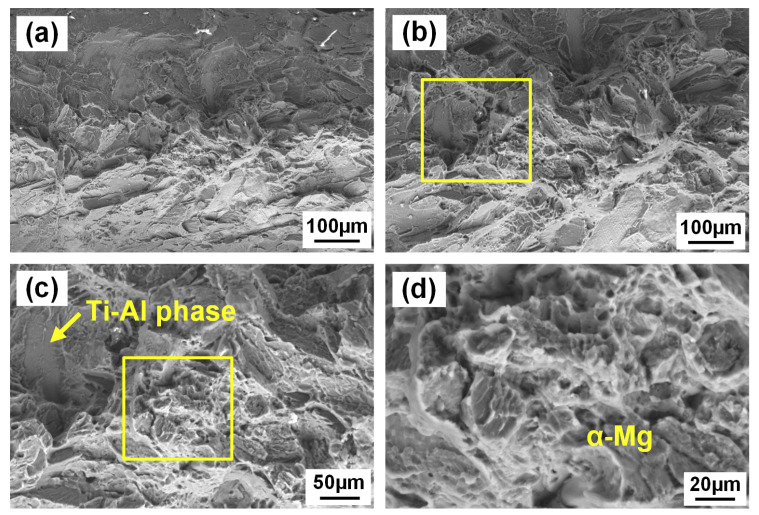
SEM morphology of fracture surface of AZ91/TA2 at the Ti side: (**a**,**b**) middle area in fracture surface; (**c**) higher magnification of zone b; (**d**) higher magnification of zone c.

**Table 1 materials-13-02743-t001:** Chemical compositions of base metals (wt.%).

Materials	Al	Zn	Mn	Fe	Si	Mg	Ti
AZ31B AZ61A AZ91D TA2	3.00 6.40 9.17 --	0.88 0.70 0.65 --	0.58 0.24 0.28 --	0.018 0.003 0.002 0.070	0.015 0.030 0.020 --	Bal. Bal. Bal. --	-- -- -- Bal.

**Table 2 materials-13-02743-t002:** Welding parameters used in present work.

Experimental Parameters	Value
Mg base metal Laser power, W Defocused distance, mm Welding speed, m/min Shielding gas flow rate, L/min Laser offset to Mg side, mm	AZ31B/AZ61A/AZ91D 2500,3000,3500,4000,4500,5000 −1.5 1.5 15 0.3

**Table 3 materials-13-02743-t003:** Parameters adopted for Mg, Al and Ti elements in Formulas (4)–(6).

Elements	*T_m_*/K	*n_ws_*/d.u.	*φ*/V	*u*	*V*/cm^3^	*R*/P
Mg	922	1.6	3.45	0.1	14	0.4
Al	933.6	2.7	4.2	0.07	10	1.9
Ti	1933	3.51	3.8	0.04	10.58	1

**Table 4 materials-13-02743-t004:** Energy-dispersive X-ray spectrometry (EDS) analysis results of points in Figure 6 (at%).

Point	Mg	Al	Ti	Possible Phases
1 2 3	89.21 68.27 12.76	6.11 31.44 27.08	4.68 0.29 60.16	Mg Mg_17_Al_12_ Ti_3_Al

**Table 5 materials-13-02743-t005:** Minimum Gibbs free energy and corresponding Ti/Al atom ratio in 0.001 Mg–Al–Ti systems.

Temperature/K	Without Limit of Al Content		With Lmit of Al Content	
	Minimum Gibbs Free Energy (Gm, KJ/mol)	Ti/Al Atom Ratio at Minimum Gm	Minimum Gibbs Free Energy (Gm, KJ/mol)	Ti/Al Atom Ratio at Minimum Gm
300	−47.43	0.994	−39.57	2.69
900	−71.78	1.035	−65.11	2.69
1300	−98.09	0.994	−91.36	2.69
1600	−121.00	0.955	−114.31	2.69
2000	−147.14	0.917	−142.14	2.69

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
