# Peer review of "Effect of Al Content in Magnesium Alloy on Microstructure and Mechanical Properties of Laser-Welded Mg/Ti Dissimilar Joints"

_materials, 2020, doi:10.3390/ma13122743_

Round 1

Reviewer 1 Report

This manuscript has adequate structure as a scientific paper demands. It was observed and analyzed morphology, microstructure and mechanical properties of joints welded by laser deep penetration. Mg base metal was used to investigate the influence of Al element in microstructure and mechanical properties of Mg/Ti dissimilar joints.

Are you sure that alpha-Mg phase and Al-Ti phase is really placed on marked spots in Figure 12? How analytic method was used to identify the phase?

What’s more, please correct some minor parts according to the guidelines:

Line 43 - adjust the degrees of Celsius as upper index

Line 87 – specify pressure of flow rate of Argon gas

Line 150 - repair word "maximum strength" to "maximum load", because strength is defined by cross section area

Figure 11.e) - Dark blue colour is hard readable on black background. Change colour or shade

Reviewer 2 Report

The reviewed article deals with interesting issue, both, from scientific and industrial point of view. The experiment was designed properly and results were clearly presented. Nevertheless during review process some remarks should be stressed:

  1. Line 18 – this sentence is a truism.
  2. Line 89 – welding speed is rather 1.5 m/min instead 1.5 mm/min.
  3. Authors did not give a Ar flow rate, please add it.
  4. There is no information about root protection, which is important for welding of titanium.
  5.  Fig.1 – it could be reduced to the one scheme and only the different magnesium alloys marks could be added.
  6. Authors gave the information about melting point of Ti and Mg, but did not give boiling point for Mg, while in line 104 the mentioned about this. Please add the value.
  7. Line 105 – what were the optimization criteria?
  8. There is a lack of methodology of mechanical properties description (in chapter no. 2), please add it.
  9. Why Authors did not carry out the VT (visual testing) of the manufactured joints?
  10. Fig.3 – the is no information about etching medium, please add it.
  11. Line 150 - Strength is given in units MPa, not N. So rather maximum force, instead strength. Additional, how many samples were tested and where are standard deviation values?
  12. Fig. 4 – interpretation, why is such big decrease for AZ61 in compare to AZ91 alloy? Please try to explain this phenomenon.
  13. Maybe a deeper discussion for diffusion between Mg and Ti (including Al) should be carried out.
  14. The samples for tensile test are without technological undercutting, why?

Reviewer 3 Report

The work is focused on laser welding of magnesium alloys with a titanium alloy in dissimilar joints configuration. The work is focused on the performance of the weld considering different aluminium contents in magnesium alloy. The quality of the weld is determined by microstructural analysis and tensile tests.   

In the present manuscript, the novelty aspect is not clear, especially considering researches on similar topic. Is the novelty aspect the deep penetration welding or is the butt joint configuration? The aim of the work is not described and the final purpose is not underlined; which is the  industrial application of this research? The interest seems on hybrid components and it is mentioned the structural application, but it is important to specify the field of interest. The materials are magnesium and titanium; they are in contrast each other in terms of corrosion behavior, in this perspective the application and the work environment became relevant.

The paper lacks of complete description and information especially in the section “Materials and Methods”. The machines are not correctly described. For example, which are the characteristics of the welding head? They are not provided. Also the experimental conditions are incomplete. Which is the gas flow rate? Which is the beam spot dimension? Which is the experimental plan? In this work, the authors talk about two experimental plans, the first one to investigate the laser power effect; the second one on the optimized condition to test the mechanical performances. The experimental plans are not correctly described. The information should be completed with tables on fixed and varied parameters, and also the number of replica for each combination.

Are you sure that the welding speed is 1.5 mm/min? Seems really slow.

In the section “Materials and Methods” there is no one description about sample preparation and characterization. Which are the adopted methods? Which are the instruments? Which is the number of replica for tensile tests? Which is the samples dimension and the speed of this test? Which is the employed standard for tensile tests?  

In terms of models (Miedema and Toop) there is another lack of information. Formulation and the used reference values are not provided.

The section “Results and Discussion” is confused due to the non-described experimental plans and due to a confused structure of the paper. In fact, in this section for the first time, it is possible understand that the authors perform different experiments (maybe with replica). The butt joints are firstly analyzed in terms of appearance but there is no one quantitative analysis on width as a function of the parameters. Considering Figure 3,  different geometrical aspects can be analyzed for the three different materials. The figure should be better described, because it is quite complicate understanding the meaning of red lines.

The figure 4 is referred to tensile tests. In the graph there is an error bar so which is the number of replica? They come from the same welded plates or from different replica of experimental condition?

In the subparagraph “Interfacial microstructure”, the employed method is not described. I think that the authors used an SEM, considering also the chemical evaluation by EDS technique. Nevertheless the EDS is useful in terms of comparison, but it is only qualitative. On the other hands which is the dimension of the spot used for the analysis? The particle P2 in Figure 5 seems really small for a correct detection. Why do they not perform XRD or XPS, in order to check the correct chemistry and phases? In figure 6 there is a graph from line scanning analysis, which is the instrument, which is the approach, again EDS?

In the subparagraph “bonding mechanism” the authors evaluate different temperature referred to different condition. This information should be put in a table to be much more readable. Moreover, in this section with thermodynamic calculation there is a lack of information about reference values, formulation. etc, as previously mentioned.

In the section “fracture behavior” it is possible found three condition, one for different base material, but there is no one replica, and it is not clear why the analysis on fracture surfaces are performed only on the worst cases.

Considering the overall paper, it is confused in the structure and the novelty aspects are not correctly emphasized and described. The information should be completed and also the analysis should be better described. In this form the submitted article is rejected.

Round 2

Reviewer 3 Report

The manuscript is improved nevertheless the answer about the aim of the work and the novelty aspects is not satisfactory and I suggest to improved this aspect underlining also the application as required in the previous comments.
